# Photocatalytic Evaluation and Application as a Sensor for the Toxic Atmospheres (Propane and Carbon Monoxide) of Nickel Antimonate (NiSb_2_O_6_) Powders

**DOI:** 10.3390/ma16145024

**Published:** 2023-07-16

**Authors:** Jacob Morales-Bautista, Héctor Guillén-Bonilla, Alex Guillén-Bonilla, Verónica-María Rodríguez-Betancourtt, Jorge Alberto Ramírez-Ortega, José Trinidad Guillén-Bonilla

**Affiliations:** 1Departamento de Ingeniería de Proyectos, CUCEI, Universidad de Guadalajara, Guadalajara 44410, Mexico; hector.guillen@academicos.udg.mx; 2Departamento de Ciencias Computacionales e Ingenierías, CUVALLES, Universidad de Guadalajara, Carretera Guadalajara-Ameca Km 45.5, Ameca 46600, Mexico; alex.guillen@academicos.udg.mx; 3Departamento de Química, CUCEI, Universidad de Guadalajara, Guadalajara 44410, Mexico; veronica.rbetancourtt@academicos.udg.mx; 4Departamento de Física, CUCEI, Universidad de Guadalajara, Guadalajara 44410, Mexico; jorge.rortega@academicos.udg.mx; 5Campus Guadalajara, UNITEC MÉXICO, Universidad Tecnológica de México, Calz. Lázaro Cárdenas 405, San Pedro Tlaquepaque 45559, Mexico; 6Departamento de Electro-Fotónica, CUCEI, Universidad de Guadalajara, Guadalajara 44410, Mexico; trinidad.guillen@academicos.udg.mx

**Keywords:** synthesis, NiSb_2_O_6_, gas sensor, photocatalytic properties

## Abstract

Nickel antimonate (NiSb_2_O_6_) powders were synthesized using a wet chemistry process assisted by microwave radiation and calcination from 600 to 700 °C to evaluate their photocatalytic and gas-sensing properties. The crystalline phase obtained at 800 °C of trirutile-type nickel antimonate was confirmed with powder X-ray diffraction. The morphology and size of the nanostructures were analyzed employing electron microscopy (SEM and TEM), identifying irregular particles and microrods (~277 nm, made up of polyhedral shapes of size ~65 nm), nanorods with an average length of ~77 nm, and nanostructures of polyhedral type of different sizes. UV-vis analysis determined that the bandgap of the powders obtained at 800 °C was ~3.2 eV. The gas sensing tests obtained a maximum response of ~5 for CO (300 ppm) at 300 °C and ~10 for C_3_H_8_ (500 ppm) at 300 °C. According to these results, we consider that NiSb_2_O_6_ can be applied as a gas sensor. On the other hand, the photocatalytic properties of the antimonate were examined by monitoring the discoloration of malachite green (MG) at five ppm. MG concentration monitoring was carried out using UV-visible spectroscopy, and 85% discoloration was achieved after 200 min of photocatalytic reaction.

## 1. Introduction

The increase in toxic gas emissions, such as CO, NO_2_, SO_2_, and O_3_, seriously threatens public health because 90% of the world’s population is exposed to polluted air, resulting in an estimated 7 million deaths yearly due to air pollution [1]. On the other hand, these atmospheric pollutants cause what is known as acid rain. These pollutants are washed away by rain and seriously contaminate soils and water sources [2].

Due to the harmful effects of polluted air, different gas detection materials have been applied to monitor and identify the toxic gases in the atmosphere. Among them, binary metal oxide semiconductors (MO_x_), such as TiO_2_, SnO, and ZnO, have been widely used. However, those binary sensors have several limitations, such as low stability under humidity, low selectivity, and high operating temperatures [3,4]. In the search for new materials with better gas detection characteristics, ternary semiconductor oxides, such as delafossites [5], perovskites [6,7], ilmenites [8], spinels [9], and trirutiles [10], have recently been studied. Among the latter, the antimonates MSb_2_O_6_ (where M = Mg, Co, Ca, Zn, Ni, or Mn) have gained much attention due to their promising characteristics as gas sensors [10,11,12]. Particularly, NiSb_2_O_6_ has been shown to have interesting properties as a gas sensor. Singh et al. [13] synthesized NiSb_2_O_6_ thin films to detect LPG and CO_2_ gases at very low operating temperatures, and Rodríguez-Betancourtt et al. [14] obtained NiSb_2_O_6_ pellets highly sensitive to C_3_H_8_ and CO atmospheres. However, there is still a lack of studies on the application of this material as a gas sensor.

On the other hand, pollutants present in the atmosphere can end up in water sources, aggravating contamination problems. To mitigate the harmful effects of pollutants, nanotechnology has gained much interest in environmental remediation applications [15,16]. In this regard, trirutile-type antimonates have shown good photocatalytic characteristics. For example, Arunkumar and Naraginti [17] studied for the first time the photocatalytic activity (under visible light conditions) of CoSb_2_O_6_, CuSb_2_O_6_, NiSb_2_O_6_, and FeSb_2_O_6_ for p-nitrophenol degradation. Liu et al. [18] studied the photocatalytic effect of ZnSb_2_O_6_ nanoparticles using the discoloration of rhodamine B and methyl orange (MO); Zhang et al. [19] synthesized nanoparticles of CaSb_2_O_6_ that showed high photocatalytic performance for the degradation of MO; Sunku et al. [20] studied the degradation of MO by using MnSb_2_O_6_ under visible light conditions; and Papi et al. [21] synthesized NiSb_2_O_6_ nanoparticles with high photocatalytic performance in the degradation of an aqueous solution of malachite green (MG). However, there have been few studies on trirutile-type antimonates applied in photocatalysis. Therefore, in this work, we evaluated the properties of NiSb_2_O_6_ as a gas sensor and photocatalyst, as this material has the potential to be applied for both air quality monitoring and environmental remediation.

The NiSb_2_O_6_ was synthesized using a microwave-assisted colloidal method. XRD microscopy was used to study its structural properties, and TEM and SEM microscopy were employed to analyze its morphological characteristics. The gas sensing response of NiSb_2_O_6_ nanoparticles was evaluated by measuring the variation in electrical resistance at different operating temperatures and concentrations of CO and C_3_H_8_. The photocatalytic performance of NiSb_2_O_6_ nanoparticles was evaluated by monitoring the discoloration of a 5 ppm aqueous solution of malachite green (MG).

## 2. Materials and Methods

### 2.1. Synthesis of NiSb_2_O_6_ Powders

The nickel antimonate synthesis was conducted using a wet chemistry method reported in [22,23]. A stoichiometric 2:1 molar ratio of nickel nitrate hexahydrate and antimony trichloride was used. A total of 1.45 g of Ni(NO_3_)_2_ꞏ6H_2_O, 2.80 g of SbCl_3_, and 2 mL of ethylenediamine (C_2_H_8_N_2_) were separately used; 5 mL of ethanol was added to the first two reactants, while 10 mL of ethanol was added to the C_2_H_8_N_2_. Each solution was kept under constant stirring at 300 rpm for 20 min at room temperature to homogenize the mixtures. After 20 min of stirring, the Ni(NO_3_)_2_ꞏ6H_2_O solution was added to the C_2_H_8_N_2_ solution. Finally, the SbCl_3_ solution was added to the mixture of Ni(NO_3_)_2_ꞏ6H_2_O and C_2_H_8_N_2_. The resulting solution was constantly stirred for 24 h at 25 °C. After completion of the stirring time, an evaporation process of the ethanol was carried out using microwave radiation, employing a General Electric ST0912C01352 domestic oven at a power of 70 W. The microwave radiation was applied in intervals of 60 s until a paste was formed, which was dried at 200 °C for 8 h and calcined at 600, 700, and 800 °C using a Novatech (Scottsdale, AZ, USA) muffle at a heating rate of 100 °C/h.

### 2.2. Physical Characterization

The crystallographic features of the NiSb_2_O_6_ calcined at different temperatures were studied using a Panalytical Empyrean diffractometer (Monterrey, Mexico) with Cuα radiation, a wavelength of 1.5406 Å, and a scan range of 10 to 90° at a rate of 0.02° per second. The band gap value of the material calcined at 800 °C was determined using UV-visible spectroscopy, applying a shift in the 200 to 600 nm range. A JEOL model JSM-6390LV scanning electron microscope (SEM, Mexico City, Mexico) in high vacuum mode was used for a detailed analysis of the compound’s microstructure, and a JEOL model JEM-2010 (Mexico City, Mexico) transmission electron microscope (TEM) with an acceleration voltage of 100 kV was employed to determine the average size of the material’s nanoparticles. For the TEM analysis, the powders were deposited in a container with 1 mL of ethyl alcohol to disperse them with ultrasound. Then, the dispersed material was poured dropwise onto a Formvar-coated copper microgrid.

### 2.3. Gas Sensing Test

To evaluate the nickel antimonate powders’ gas detection ability, pellets of 15 mm in diameter and 0.5 mm in thickness were made from the powders using a hydraulic press (Equip-25 Ton, Mexico City, Mexico) at a pressure of 10 Ton for 15 min; 0.4 g of NiSb_2_O_6_ calcined at 800 °C were used for this purpose. Subsequently, two ohmic contacts were attached to the pellets using colloidal silver paint (Alfa Aesar, Ward Hill, MA, USA 99%) to improve the contact between the pellets’ surface and the measurement system’s electrodes. The procedure to carry out the measurements is the following: firstly, the sample is placed in a 12 L chamber with ambient air, then a low vacuum is created in the chamber to approximate a 0 ppm concentration, then the concentration of the test gas to be analyzed is injected. The sample reacts after 50 s due to oxygen species adsorbed on the sample surface, the measurement is taken after 10 min in order to have a more stable measurement of the resistance change. It is important to mention that the equipment is designed for static measurements, that is, the response is affected exclusively by the test gas, for this reason there is no stabilizer gas flow, such as air.

The NiSb_2_O_6_ pellets were installed in the measurement chamber with a vacuum capacity of 10^−3^ Torr (Figure 1), and measurements were made in atmospheres of C_3_H_8_ and CO. For this, concentrations of 1, 5, 50, 100, 200, 300, 400, and 500 ppm of C_3_H_8_, and 1, 5, 50, 100, 200, and 300 ppm of CO were injected into the system. The operating temperatures were 100, 200, and 300 °C. The gas concentrations were controlled using a Leybold detector. The operating temperature was monitored using a highly sensitive type K thermocouple. The pellets’ varielectrical resistance was recorded using a Keithley 2001 digital multimeter (Beaverton, OR, USA). The pellets’ response was calculated using the equation reported in [24,25]:(1)$$S=\frac{G_{g}-G_{0}}{G_{0}}$$
where G_g_ and G_0_ are the pellets’ conductances (i.e., the reciprocal of the resistance) in the presence of gas (CO or C_3_H_8_) and air, respectively.

### 2.4. Photocatalytic Activity Test

To analyze their photocatalytic activity, 0.25 g of NiSb_2_O_6_ powders were pressed into pellets (Ø 5 mm) and suspended in a quartz cell containing 3.5 mL of a 5 ppm malachite green aqueous solution. The suspension was kept in the dark for 30 min to reach the adsorption equilibrium. Then the suspension was put into an annular photoreactor equipped with a 15 W UV-254 nm lamp (Figure 2) and irradiated for 200 min. The discoloration was monitored at 30 min intervals using a JASCO V-670 UV-vis spectrophotometer.

## 3. Results

### 3.1. XRD Analysis

Figure 3 shows diffractograms of the NiSb_2_O_6_ calcined at 600, 700, and 800 °C. According to PDF No. 38-1083, diffraction peaks corresponding to the NiSb_2_O_6_’s crystalline phase were identified at 2θ = 19.22°, 21.43°, 27.14°, 33.50°, 34.99°, 38.80°, 40.19°, 44.73°, 53.22°, 56.03°, 60.23°, 62.77°, 63.32°, 67.20°, 67.78°, 73.89°, 80.89°, and 86.83°. However, a small secondary phase associated with NiO was observed at 2θ = 36.82° (PDF No. 44-1159) and 62.76° (PDF No. 44-1159). Comparing the diffractograms with PDF No. 38-1083, the NiSb_2_O_6_ crystallized in a tetragonal trirutile-type crystal structure [22] with cell parameters a = 4.641 Å and c = 9.223 Å, and space group P42/mnm [22,26]. Figure 3 shows pronounced broadening as well as the high intensity of NiSb_2_O_6_’s peaks. It has been reported in the literature that both features are indicative of a crystalline nanometric size [27]. To determine the crystallite size, Scherrer’s equation [27] was used:(2)$$t=\frac{0.9\lambda }{\beta cos\theta }$$
where *λ* is the radiation’s wavelength (Cu = 1.5406 Å), *β* is the peak width measured at half of the maximum intensity, and *θ* is Bragg’s angle. The most intense peak (110) corresponding to the last calcination (800 °C, see Figure 3) was considered for the calculation. A crystallite size of ~43.48 nm was obtained.

After extensive literature research, we compared our results with those of other groups that have prepared the same compound and found that, in our case, we obtained the NiSb_2_O_6_ crystalline phase at a lower calcination time (<5 h) from 600 to 800 °C. In the references [28,29], NiSb_2_O_6_’s crystalline phase formation was reported at a residence time of 3 days at 800 °C. Some authors used the solid-state reaction process (ceramic method), which required several days to obtain the crystalline phase of NiSb_2_O_6_, CoSb_2_O_6_, and CuSb_2_O_6_ [28]. In our case, we used a microwave-assisted wet chemistry method, which is economical, simple, and with easy control of chemical reactions and physical processes (such as low temperature to obtain the crystalline phase).

### 3.2. SEM Analysis

Figure 4 shows SEM micrographs of powders of the NiSb_2_O_6_ calcined at 800 °C. Magnifications used for the material’s microstructural analysis were 6.0, 10.0, 15.0, 60.0, and 70.0 kx. Based on the images, it was found that the material’s entire surface is composed of particles in the form of rods, polyhedra, and other irregular shapes. Figure 4a–c show a high agglomeration of polyhedral particles and other particles forming rods of different lengths. These particles agglomerated in such a way that they formed irregular structures of varying sizes (Figure 4d–f). In Figure 4d,f, the growth of smaller microrods can be observed, which had a common point of origin but grew in different directions. This formation of microrods is mainly attributed to the fact that, during the thermal treatment, the growth of very fine particles is favored, as well as their subsequent agglomeration, resulting in the formation of characteristic morphologies as shown. In addition, the residence time of the material (5 h) at 800 °C played a significant role in the morphologies’ growth.

The size of the microrods was estimated in the range of 100 to 500 nm, with an average of ~277 nm and a standard deviation of ~±83 nm (Figure 5a), while the size of the other particles was estimated in the range of 20 to 130 nm, with an average of ~65 nm and a standard deviation of ~±21 nm (Figure 5b). Multiple SEM micrographs, where the particles were clearly identifiable, were considered for the microstructures’ size calculation. Several authors have reported that the use of chelating agents such as ethylenediamine in the synthesis promotes the formation of such microstructures. This is because ethylenediamine reacts with temperature to form a mesh that captures metal particles, which agglomerate as the calcination temperature increases, resulting in the growth of rods and other morphologies [30]. Studies have also suggested that the use of ethylenediamine can lead to the formation of diverse nanostructures including nanoparticles, nanorods, nanowires, porous particles, and hexagonal nanostructures [31,32]. The microstructures in our study followed the crystallization principles established by LaMer and Dinegar [33]. According to these principles, nuclei formed due to chemical reactions until a saturation limit was reached, after which the particles grew until the system’s solubility reached equilibrium. This condition favored the production of morphologies such as those shown in Figure 4d–f.

### 3.3. TEM Analysis

To further examine the microstructure of the nickel antimonate calcined at 800 °C, transmission electron microscopy (TEM) in imaging mode was used. The resulting images, shown in Figure 6, reveal mainly nanorods and other particle types. This analysis confirmed the SEM results, which identified rods of different sizes and irregularly shaped nanoparticles measuring ~90 nm. Their thickness caused dark areas in the nanostructures, which impeded electron transmission. The nanorods and nanoparticles seen in Figure 6a–f agglomerated together due to the effect of the calcination temperature and ethylenediamine. These particles varied in shape and size and had small growth nuclei that caused the nanorods to grow in different directions [34]. The nanorod sizes ranged from 40 to 180 nm (Figure 6b–f), with a mean size of ~77 nm and a standard deviation of ±34 nm (see Figure 7).

Agreeing with other authors [35], ethylenediamine helps to achieve specific sizes and morphologies, such as nanorods, microrods, nanowires, and polyhedra. In our case, we obtained nanorods and nanoparticles (Figure 6a–f) using ethylenediamine and an alternative synthesis process (wet chemistry), which allowed us to control the compound’s structure in short calcination times (5 h).

### 3.4. UV-Vis Analysis

To identify the characteristic absorbance bands and determine the value of the forbidden band of the oxide calcined at 800 °C, UV-vis spectroscopy was employed. In Figure 8, absorbance is shown as a function of the compound’s wavelength. In the 200–500 nm range, characteristic absorption bands of a material belonging to the family of semiconductor oxides with a trirutile-type structure were identified [13,36,37]. On the other hand, in the 600–800 nm range, bands associated with the oxygen-metal bond (in our case, Ni-O) were identified [38,39].

Tauc’s equation [40] was employed to calculate the value of the band gap. Previous studies have indicated that materials like the NiSb_2_O_6_ exhibit direct transitions between their energy bands (*n* = ½) [13,29]. In Figure 8, (*α*ℎ*υ*)^2^ was plotted against energy, and the resulting spectrum was fitted to a line, yielding a forbidden band value of 3.2 eV. In reference [13], the band gap was reported as 3.92 eV. Another study [29] reported values in the range of 2.6–2.8 eV for a material calcined at 700 °C and 2.8–2.9 eV for a material calcined at 900 °C. It should be noted that the forbidden band value is strongly influenced by the synthesis method and the heat treatment employed, as mentioned by several authors [13,22,29], which is consistent with the findings of this study.

### 3.5. Static Tests in CO and C_3_H_8_

To check nickel antimonate’s detection ability to different gas concentrations, its pellets were subjected to 1–300 ppm of CO and 1–500 ppm of C_3_H_8_ at temperatures 100, 200, and 300 °C. The results are shown in Figure 9 and Figure 10 as a function of the test gas concentration and the operating temperature. In the case of CO (Figure 9a,b), the material’s response increased as the concentration and operating temperature rose. This increase is mainly attributed to oxygen species (like $O_{2}^{-}$, $O^{-}$, or $O^{2-}$) on the surface of the pellets, which rapidly reacted when the CO was injected into the measurement chamber. Furthermore, the excellent oxide response is associated with higher oxygen desorption due to the operating temperature [23,24]. When the pellet was at 100 and 200 °C, a slight increase in gas sensing response was recorded: ~0 and ~1 at 300 ppm of CO, respectively. This low response was because the thermal energy was insufficient for the oxygen species to react on the pellets’ surface [41], which induced poor kinetic activity between them, the material’s particles, and the CO molecules [42,43]. However, when the temperature was raised to 300 °C, values of 0, 0, 1, 2, 4, and 5 were recorded, corresponding to 1, 5, 50, 100, 200, and 300 ppm of CO. The maximum response at 300 °C was ~5 at 300 ppm of CO. The excellent response is mainly attributed to the fact that the oxygen species $O^{-}$ and $O^{2-}$ are highly reactive at temperatures greater than 150 °C compared to those species ($O_{2}^{-}$) that appear below that temperature [4,44]. It has been reported that an increase in the operating temperature favors a high interaction between the pellets’ surface with the CO molecules [41], thereby provoking variation in electrical resistance and, therefore, an increase in response, as occurred in our case (Figure 9a,b).

In Figure 10, the results of the tests in C_3_H_8_ atmospheres at 100, 200, and 300 °C are depicted. As in the previous case, when injecting the gas into the measurement chamber, the pellets showed varied electrical resistance due to temperature and C_3_H_8_ concentrations. However, when the gas was removed from the chamber, the pellets’ electrical resistance returned to their initial values (i.e., before starting the tests), thus corroborating the reproducibility of the experiments. According to Figure 10, the response magnitude rose significantly as the C_3_H_8_ concentration (from 1 to 500 ppm) and the operating temperature (from 100 to 300 °C) increased. Thus, at 100 and 200 °C and 500 ppm of C_3_H_8_, small response values were estimated: ~1 and ~2, respectively. The maximum response recorded was ~10 at 500 ppm of C_3_H_8_ and 300 °C. This trend in the increase in response is attributed to the fact that when the gas was injected into the chamber, the C_3_H_8_ molecules diffused on the pellets’ surface [45], reacting with the oxygen available ($O^{-}$ and $O^{2-}$) due to the temperature effect [23,46]. According to the literature, the different oxygen species ($O_{2}^{-},O^{-}$, or $O^{2-}$) that reacted at temperatures close to 300 °C [4,44] are responsible for the variation in the electrical resistance and, consequently, the increased NiSb_2_O_6_’s response in C_3_H_8_ atmospheres. The reaction between the test gas and the oxygen species caused a high gas–solid interaction [37,44,46], causing the compound’s increased response. A possible chemical reaction between materials such as the one studied here and C_3_H_8_ has been proposed: C_3_H_8_ + 10O_2_^−^ → 3CO_2_ + 4H_2_O + 10e^−^ [46]. This means that when the pellets’ surface comes into contact with the C_3_H_8_, the latter dissociates before reacting with the ionosorbed oxygen species [24,46], causing a redox reaction that provokes changes in electrical resistance and, as a result, an increase in NiSb_2_O_6_’s response (Figure 10a,b).

### 3.6. Photocatalytic Tests

NiSb_2_O_6_’s photocatalytic activity was monitored with the discoloration of malachite green (MG). The evolution of the photo discoloration and the discoloration percentage are shown in Figure 11a,b, respectively. According to Figure 11b, after being in the dark for 30 min, the sample absorbed 30% of the dye. However, MG’s concentration decreased significantly during the UV light irradiation, which meant that when the NiSb_2_O_6_ particles acted as a photocatalyst, up to 85% of MG’s photo discoloration occurred after 200 min of irradiation. These results are competitive according to the literature. Table 1 shows the photocatalytic performance of different antimonates with a trirutile structure.

The reaction kinetics of the photo discoloration was estimated using the first-order Langmuir–Hinshelwood equation [47]:(3)$$\ln\frac{C_{0}}{C}=kt$$
where *C*_0_ is MG’s concentration before the photo discoloration, *C* is the concentration at the beginning of the photocatalytic reaction, *t* is the reaction time, and *k* is the reaction rate constant [47]. The latter was estimated using the slope of Figure 11c (=0.0069).

The good photocatalytic performance of NiSb_2_O_6_ pellets can be attributed to the nanorods, since according to the literature, nanostructures such nanorods have shown to be efficient to enhance photocatalytic performance. C. J. Chang et al. [48] studied the effect of nanorod-type nanostructures on the surface of ZnO films, concluding that the photocatalytic performance increases with the presence of these structures due to their higher surface area. S. Zhang et al. [49] prepared TiO_2_, nanorods, TiO_2_ nanosheets, and TiO_2_ nanospheres; the best photocatalytic activity occurred in the nanorods due not only to the surface area, but also because this type of structures can provide direct conduction paths for photo-induced carriers, resulting in better photocatalytic activity. Y. C. Chang et al. [50] synthesized sodium titanate nanorods that showed good photocatalytic activity. The better photocatalytic performance may be due to the higher amounts of oxygen vacancies that can occur in this type of structure, which act as electron traps, decreasing recombination and then improving the photocatalytic performance. Therefore, various phenomena can occur in this type of structures that may contribute to improving the photocatalytic activity.

Respecting the photocatalysis mechanism shown in Equations (4)–(7), it has been reported that the photon absorption by the photocatalyst leads to the excitation of electrons from the valence to the conduction bands, generating electron–hole (e^−^-h^+^) pairs (Equation (4)). The holes react with adsorbed water molecules on the photocatalyst’s surface (Equation (5)), producing hydroxyl radicals (OH^●^). Electrons react with dissolved oxygen molecules in the water (Equation (6)), producing superoxide radicals (O_2_^−●^) [21,51,52]. Therefore, the MG may photodegrade by reacting with hydroxyl or superoxide radicals (Equation (7)).

The OH^●^ generated is key in the photocatalytic process, since it reacts with the substances adsorbed on the surface and degrades them.
(4)$$Ni{Sb}_{2}O_{6}+hv\to e^{-}+h^{+}$$
(5)$$H_{2}O+h^{+}\to {OH}^{•}+H^{+}$$
(6)$$O_{2}+e^{-}\to O_{2}^{-}$$
(7)$$Dye\left(MG\right)+{OH}^{•}\left(or\;O_{2}^{•-}\right)\to degradated\;products$$

## 4. Conclusions

Applying a wet-chemistry method assisted with microwave radiation favored obtaining the crystalline phase of nickel antimonate (NiSb_2_O_6_) at a temperature of 800 °C in just five hours. SEM studies of the NiSb_2_O_6_’s microstructure allowed us to identify morphologies in the form of microrods, polyhedrons, and other particles with no apparent shape. With TEM, it was estimated that the average size of the nanorods was ~77 nm. UV-vis spectroscopy estimated a value of the NiSb_2_O_6_’s band gap of ~3.2 eV. This value is within the energy parameters for antimonates with a trirutile-type structure. NiSb_2_O_6_ pellets showed excellent responses of ~5 at 300 ppm of CO and ~10 at 500 ppm of C_3_H_8_, at 300 °C. These responses are attributed to the microstructure obtained during the synthesis. The particles’ morphological characteristics and size were closely related to the changes in electrical resistance and, therefore, to the increase in the compound’s response. The photocatalytic activity of NiSb_2_O_6_ nanostructures reached 85% degradation of malachite green. This suggests that NiSb_2_O_6_ is a potential photocatalytic material. However, it is a material little studied for applications as a gas sensor or photocatalyst. To encourage the study of this material, it is recommended to use our synthesis method, which is easy to carry out, relatively cheap, and does not require sophisticated equipment to control the physical and chemical parameters involved in the oxide’s preparation.

## Figures and Tables

**Figure 1 materials-16-05024-f001:**
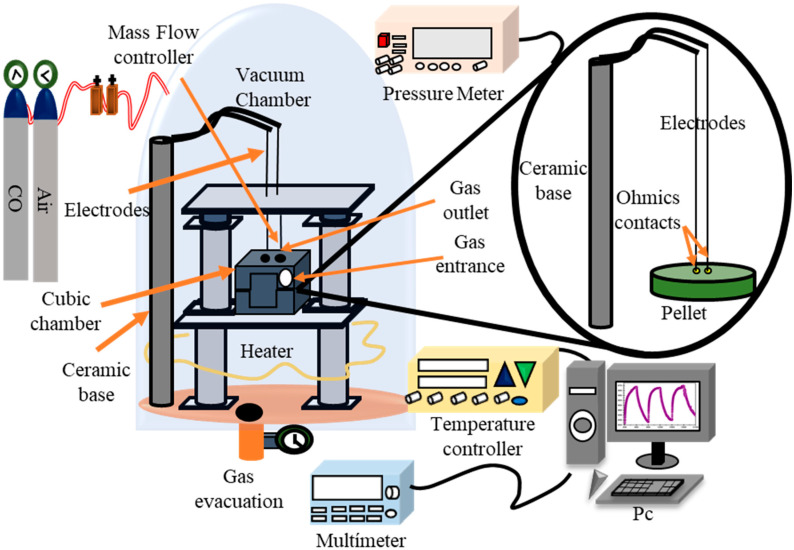
Diagram of the detection system of controlled CO and C_3_H_8_ atmospheres at different operating temperatures.

**Figure 2 materials-16-05024-f002:**
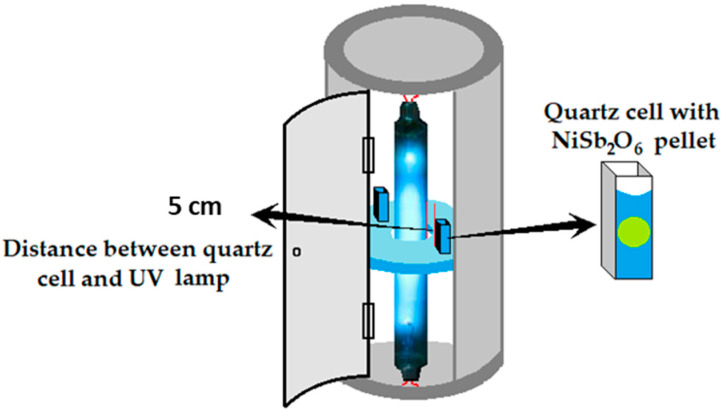
Annular reactor for the photocatalytic tests of powders of NiSb_2_O_6_ calcined at 800 °C.

**Figure 3 materials-16-05024-f003:**
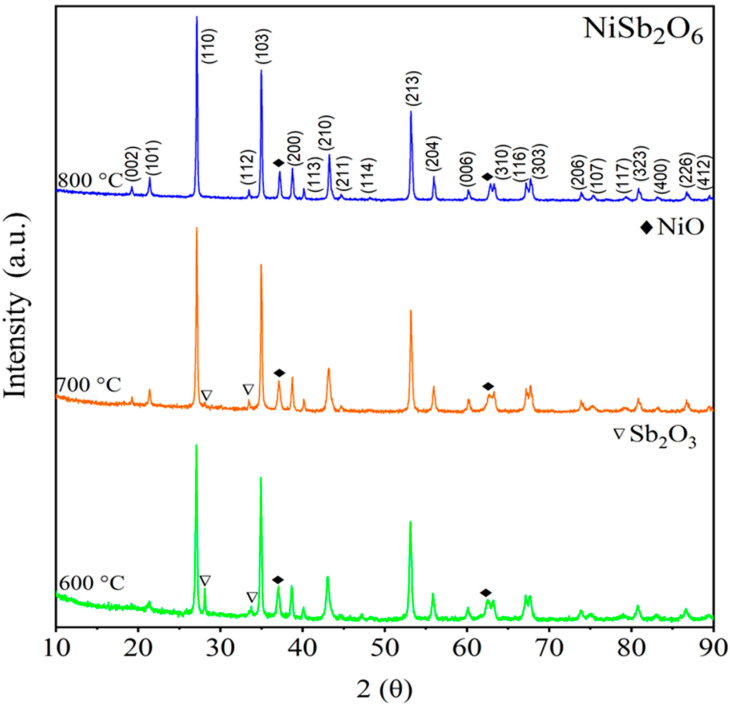
X-ray diffraction of NiSb_2_O_6_ calcined in air at 600, 700, and 800 °C.

**Figure 4 materials-16-05024-f004:**
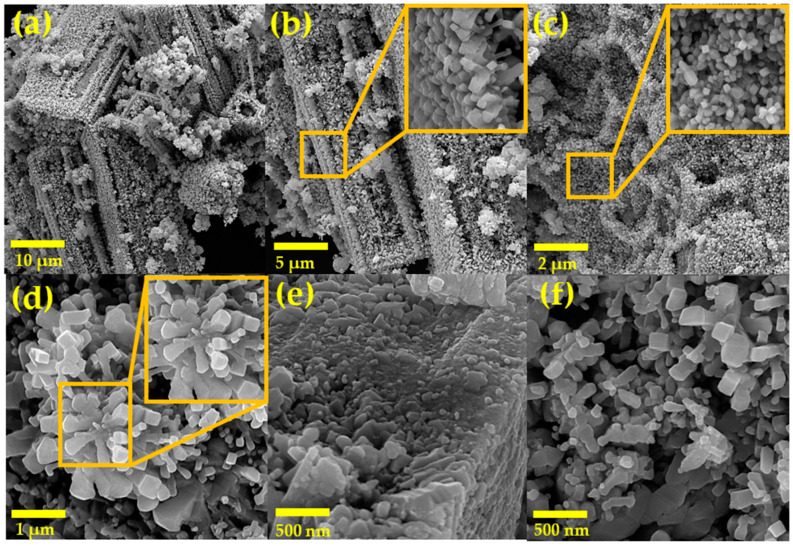
SEM micrographs of powders of NiSb_2_O_6_ calcined at 800 °C at magnifications: (**a**) 6.0 kx, (**b**) 10.0 kx, (**c**) 15.0 kx, (**d**) 60.0 kx, (**e**) 70.0 kx, and (**f**) 70.0 kx.

**Figure 5 materials-16-05024-f005:**
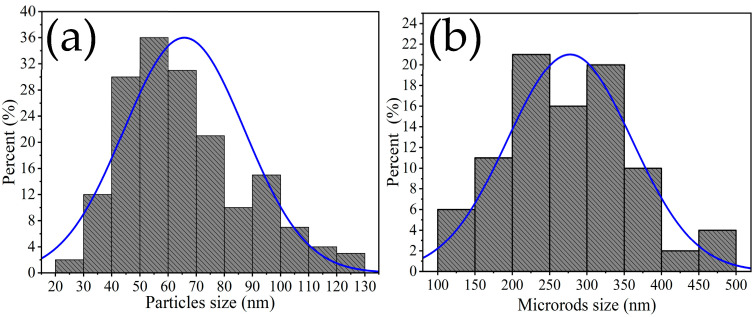
Particle size distribution: (**a**) bars, (**b**) other types of particles.

**Figure 6 materials-16-05024-f006:**
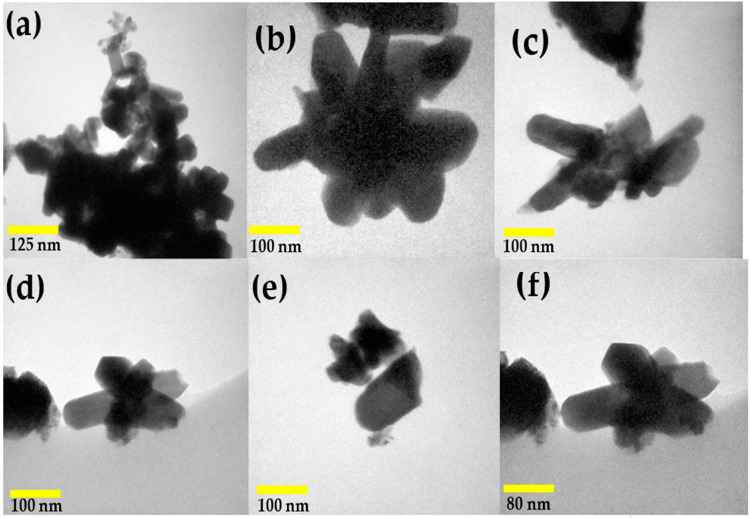
TEM micrographs of (**a**) nanoparticles and (**b**–**f**) nanorods of NiSb_2_O_6_ calcined at 800 °C.

**Figure 7 materials-16-05024-f007:**
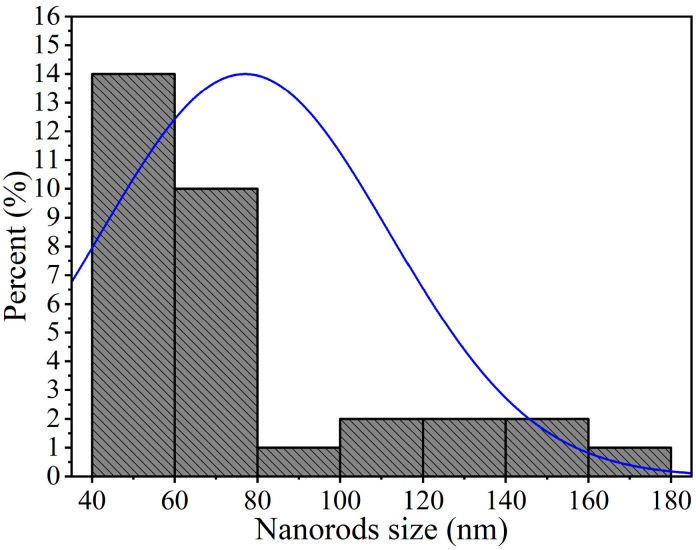
Size distribution of nanorods of NiSb_2_O_6_ calcined at 800 °C.

**Figure 8 materials-16-05024-f008:**
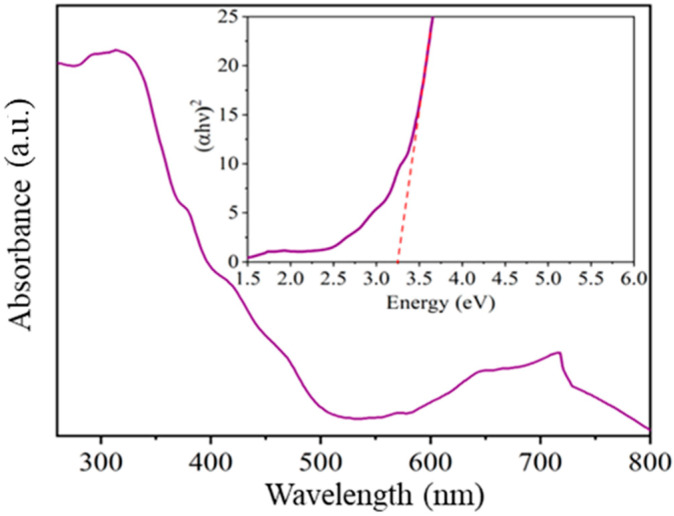
Band gap value determined by the UV-Vis spectroscopy of NiSb_2_O_6_ calcined at 800 °C.

**Figure 9 materials-16-05024-f009:**
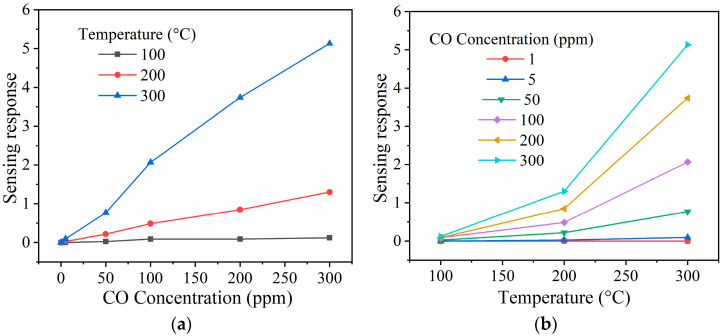
Sensing response of the NiSb_2_O_6_ pellets as a function of (**a**) the concentration of CO and (**b**) the operating temperature.

**Figure 10 materials-16-05024-f010:**
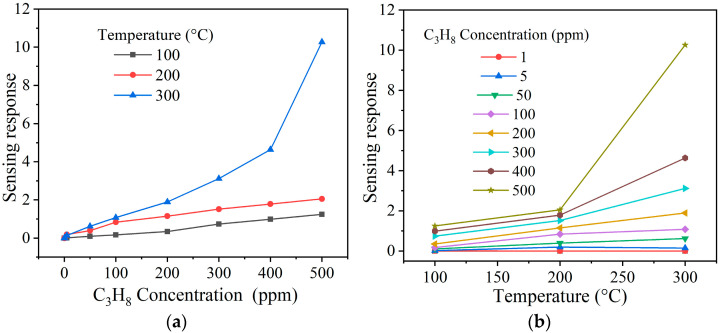
Sensing response of the NiSb_2_O_6_ pellets as a function of (**a**) the concentration of C_3_H_8_ and (**b**) the operating temperature.

**Figure 11 materials-16-05024-f011:**
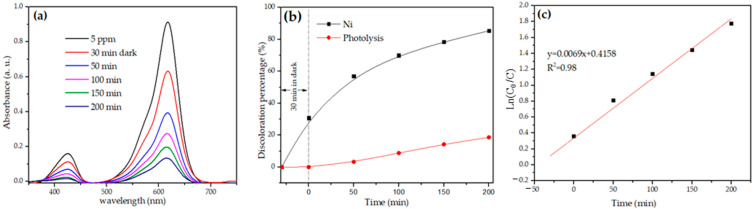
(**a**) Evolution of MG’s photo discoloration, (**b**) MG’s discoloration percentage, (**c**) MG’s first-order kinetic photodegradation.

**Table 1 materials-16-05024-t001:** Photocatalitic performance of different trirutile antimonates.

Trirutile Semiconductor	Analyte	Photocatalytic Performance (%)	Reference
CaSb_2_O_6_	Methyl orange	94	19
CoSb_2_O_6_	Acridine orange	61	54
MgSb_2_O_6_	P-bromophenol	89	53
ZnSb_2_O_6_	Methyl orange	75	18
MnSb_2_O_6_	Methyl orange	85	20
CuSb_2_O_6_	Rhodamine 6 g	96	55
NiSb_2_O_6_	Malachite green	96	21

## Data Availability

The data that support the findings of this study are available from the corresponding authors upon request.

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
