# Peer review of "Photocatalytic Evaluation and Application as a Sensor for the Toxic Atmospheres (Propane and Carbon Monoxide) of Nickel Antimonate (NiSb2O6) Powders"

_materials, 2023, doi:10.3390/ma16145024_

Round 1

Reviewer 1 Report

This paper focuses on synthesizing and characterizing NiSb2O6 powders, demonstrating their potential as a gas sensor for C3H8 and CO atmospheres. The findings also highlight the material's effective photocatalytic properties, as evidenced by the significant discoloration of malachite green. These results indicate the suitability of NiSb2O6 for gas sensing and environmental remediation applications. Overall, the writing is clear and thorough; the paragraph has some minor typo issues. 

1.     Some sentences may miss from 149 line to 160 line.

2.     Figure 2 has some letters on the left bottom part.

3.     The figure caption in Figure 6 is not clear. Which image is for nanorods, and which image is for nanoparticles?

4.     Figure 8, the absorbance unit is (a.u.), not (u.a.)

5.     Figure 11(a), the absorbance unit is missing.

Author Response

General Editor
Materials MDPI

July 05, 2023

Subject: Response to comments on paper ID: materials-2456064

We thank the reviewers for the valuable comments about our article, which helped us to improve it. So, we are pleased to re-submit for publication the revised version of our paper entitled “Photocatalytic Evaluation and Application as a Sensor for Toxic Atmospheres (Propane and Carbon Monoxide) of Nickel Antimonate (NiSb2O6) Powders”, which contains corrections according to all reviewer’s remarks. We hope the manuscript’s updated version meets your expectations.

Author´s Reply to the Review Report

Reviewer 1

 Comments and Suggestions for Authors.

This paper focuses on synthesizing and characterizing NiSb2O6 powders, demonstrating their potential as a gas sensor for C3H8 and CO atmospheres. The findings also highlight the material's effective photocatalytic properties, as evidenced by the significant discoloration of malachite green. These results indicate the suitability of NiSb2O6 for gas sensing and environmental remediation applications. Overall, the writing is clear and thorough; the paragraph has some minor typo issues.

Comment 1: Some sentences may miss from 149 line to 160 line.

 Response: We have corrected that part.

Comment 2: Figure 2 has some letters on the left bottom part.

 Response: Figure 2 has been modified.

Comment 3: The figure caption in Figure 6 is not clear. Which image is for nanorods, and which image is for nanoparticles?

 Response: Figure 6’s title has been modified (section “3.3 TEM”, page 8).

 Comment 4:  Figure 8, the absorbance unit is (a.u.), not (u.a.).

 Response: We modified the absorbance unit in Figure 8 (section “UV-Vis Analysis”, page 10).

 Comment 5:  Figure 11(a), the absorbance unit is missing.

Response: We added the absorbance unit in Figure 11a (section “Photocatalytic tests”, page 12).

Sincerely,

Corresponding author: Dr. Jacob Morales Bautista

Professor - Research Scientist, Department of Engineering Projects, CUCEI

University of Guadalajara

Blvd. Marcelino García Barragán #1421, 44430 Guadalajara, Jalisco, México.

+52 (33) 1378 5900 ext. 27655

jacob.mbautista@academicos.udg.mx

Reviewer 2 Report

In this manuscript triple oxide nanomaterial is developed. It can be used to ensure environmental safety when creating gas sensors and photocatalysts. The manuscript describes a method of wet chemical synthesis of nickel antimodate nanopowder using ethylenediamine as a chelating agent. The phase composition and microstructure of the nanopowder are studied. Gas sensor properties were analyzed in propane and carbon monoxide atmospheres. The photocatalytic activity was studied on the example of the decomposition of malachite green under the exposure of ultraviolet radiation. The resulting material has high potential for further enhancement of gas sensors and photocatalysts.

The manuscript has a high scientific level, but for publication it is necessary to correct the following shortcomings:

1. The word “synthesized” in the title raises questions. If this word is used, then the synthesis method should be specified, or it can be removed from the title.

2. In the explanation of formula (1), it is mentioned that G0 is the conductance of the pellet in CH3 atmosphere. However, this value corresponds to the conductance of the pellet in air atmosphere.

3. In the equation of chemical reaction in line 320, there is one electron, but there should be 10 electrons.

4. In Figure 4, the scale bar and captions are barely visible, they should be enlarged.

Author Response

General Editor
Materials MDPI

July 05, 2023

Subject: Response to comments on paper ID: materials-2456064

We thank the reviewers for the valuable comments about our article, which helped us to improve it. So, we are pleased to re-submit for publication the revised version of our paper entitled “Photocatalytic Evaluation and Application as a Sensor for Toxic Atmospheres (Propane and Carbon Monoxide) of Nickel Antimonate (NiSb2O6) Powders”, which contains corrections according to all reviewer’s remarks. We hope the manuscript’s updated version meets your expectations.

Reviewer 2

In this manuscript triple oxide nanomaterial is developed. It can be used to ensure environmental safety when creating gas sensors and photocatalysts. The manuscript describes a method of wet chemical synthesis of nickel antimodate nanopowder using ethylenediamine as a chelating agent. The phase composition and microstructure of the nanopowder are studied. Gas sensor properties were analyzed in propane and carbon monoxide atmospheres. The photocatalytic activity was studied on the example of the decomposition of malachite green under the exposure of ultraviolet radiation. The resulting material has high potential for further enhancement of gas sensors and photocatalysts.

The manuscript has a high scientific level, but for publication it is necessary to correct the following shortcomings:

Comment 1:  The word “synthesized” in the title raises questions. If this word is used, then the synthesis method should be specified, or it can be removed from the title.

 Response: We agree. We concisely modified the manuscript’s title.

 Comment 2:  In the explanation of formula (1), it is mentioned that G0 is the conductance of the pellet in C3H8 atmosphere. However, this value corresponds to the conductance of the pellet in air atmosphere.

 Response: In the revised version, the explanation was modified (section “2.3 Gas sensing test”, page 3).

Comment 3: In the equation of chemical reaction in line 320, there is one electron, but there should be 10 electrons.

 Response: In the revised version, the chemical reaction in line 320 was corrected (section “3.2 SEM analysis”, page 7).

Comment 4: In Figure 4, the scale bar and captions are barely visible, they should be enlarged.

 Response: We agree. The scale bar was added to each image (section “Static tests in CO and C3H8”, page 7).

Sincerely,

Corresponding author: Dr. Jacob Morales Bautista

Professor - Research Scientist, Department of Engineering Projects, CUCEI

University of Guadalajara

Blvd. Marcelino García Barragán #1421, 44430 Guadalajara, Jalisco, México.

+52 (33) 1378 5900 ext. 27655

jacob.mbautista@academicos.udg.mx

Reviewer 3 Report

The manuscript entitled “Photocatalytic Evaluation and Application as a Sensor for Toxic Atmospheres (Propane and Carbon Monoxide) of Synthesized Nickel 3 Antimonate (NiSb2O6) Powders” was proposed for detection of toxic gases. The author’s synthesis approach is conventional as are other techniques, but detection of toxic gases and Photocatalytic Evaluation with the proposed systems could be convincing in the respective field. I recommend that the author revise the manuscript before taking further decisions about it. Further, few comments for the author should respond to are mentioned.

1.            The author should have written write brief abstract because length is extremely extensive.

2.            The size distribution of NiSb2O6 nanorods for TEM images provided by the author is not clear to reader.

3.            The proposed system's gas detecting method is not clear to the reader, so the author needs to improve the mechanism with proper description and corresponding references. More specifically, how do the NiSb2O6 nanorods affect the photocatalytic evaluation? 

4.            Check the absorption unites in Fig. 8

5.            Check the lines no; 92, 96, 97, 98 and 306. Similarly, numerous typos have been noticed in the manuscript. Before sending the manuscript to the appropriate journal, the author should double-check it.

6.            The author should compare the obtained results with formerly published articles.

7.            The author needed to ensure stability for the proposed system.

8.            Add references that are published in recent times in the introduction.

The author's English in the manuscript needs to be improved. 

Author Response

General Editor
Materials MDPI

July 05, 2023

Subject: Response to comments on paper ID: materials-2456064

We thank the reviewers for the valuable comments about our article, which helped us to improve it. So, we are pleased to re-submit for publication the revised version of our paper entitled “Photocatalytic Evaluation and Application as a Sensor for Toxic Atmospheres (Propane and Carbon Monoxide) of Nickel Antimonate (NiSb2O6) Powders”, which contains corrections according to all reviewer’s remarks. We hope the manuscript’s updated version meets your expectations.

Reviewer 3

 The manuscript entitled “Photocatalytic Evaluation and Application as a Sensor for Toxic Atmospheres (Propane and Carbon Monoxide) of Synthesized Nickel 3 Antimonate (NiSb2O6) Powders” was proposed for detection of toxic gases. The author’s synthesis approach is conventional as are other techniques, but detection of toxic gases and Photocatalytic Evaluation with the proposed systems could be convincing in the respective field. I recommend that the author revise the manuscript before taking further decisions about it. Further, few comments for the author should respond to are mentioned.

Comment 1:  The author should have written write brief abstract because length is extremely extensive.

Response: We agree; many thanks. The abstract has been reduced to a standard length.

Comment 2:  The size distribution of NiSb2O6 nanorods for TEM images provided by the author is not clear to reader.

 Response: We appreciate your observation. The TEM image’ caption has been corrected for a better understanding (lines 239-240).

Comment 3: The proposed system's gas detecting method is not clear to the reader, so the author needs to improve the mechanism with proper description and corresponding references. More specifically, how do the NiSb2O6 nanorods affect the photocatalytic evaluation?

Response: Thanks for the comment. Nanorods have several characteristics that are beneficial for applications such as photocatalysis. Therefore, we added information on this topic based on an additional literature review (see section 3.6, page 12). Furthermore, the gas system details were added (see section 2.3, pages 2-3).

Comment 4: Check the absorption unites in Fig. 8.

 Response: Fixed (section 3.4, page 10).

Comment 5: Check the lines no; 92, 96, 97, 98 and 306. Similarly, numerous typos have been noticed in the manuscript. Before sending the manuscript to the appropriate journal, the author should double-check it.

 Response: Corrected; many thanks.

Comment 6: The author should compare the obtained results with formerly published articles.

 Response: We agree. Comparative information on the photocatalytic results was added (see section 3.6, page 12).

Comment 7: The author needed to ensure stability for the proposed system.

 Response: Details about the measurement process for the sensing system were added (see section 2.3, pages 2-3).

Comment 8: Add references that are published in recent times in the introduction.

 Response: We added up-to-date references on these ternary materials (see references 7-8 and 17).

Sincerely,

Corresponding author: Dr. Jacob Morales Bautista

Professor - Research Scientist, Department of Engineering Projects, CUCEI

University of Guadalajara

Blvd. Marcelino García Barragán #1421, 44430 Guadalajara, Jalisco, México.

+52 (33) 1378 5900 ext. 27655

jacob.mbautista@academicos.udg.mx

Reviewer 4 Report

The article describes the method of obtaining the material, provides its characterization and the results of the study of sensory properties. The synthesis is well described and the characterization is well done. A few notes:

1) It is unclear why antimony chloride rather than nitrate was chosen as the precursor. The use of chlorides in the production of sensory materials tends to be avoided, since they remain in the final material and contribute significantly to its electrical conductivity, which reduces the sensory response. Nitrates thermally decompose, so they are preferred for obtaining sensory materials.

2) In sensory journals, authors usually show "raw", raw data that shows the drift of the background resistance of the sample, the time to reach stationary values ​​after the injection of the target gas, the relaxation time after replacing the target gas with air. The article would have won with such data.

3) In scientific research, not only the result itself is important, but also the number of significant figures in this result, which indicate the accuracy of the method. The authors present the size of nanorods with an accuracy of three significant figures (77.0 nm), however, electron microscopy methods do not allow one to determine the size with such accuracy - two significant figures would be enough. It is even more strange to see the value of the standard deviation, for which four significant digits of 34.82 nm are given, although the standard deviation should contain as many digits after the decimal point as the value whose error it characterizes.

A similar remark should be made about the results of sensory response. First, it is incorrect to refer to sensory response as sensitivity in Fig. 9, 10 (this error, unfortunately, occurs in many papers). Sensitivity is the derivative of the response with respect to concentration. Secondly, four significant digits for a sensory response is unreasonably large, since both the background resistance value always drifts, and the final resistance value does not remain stationary after the application of the target analyte. Two significant digits would suffice. In some cases, three significant figures can be used, but this needs justification.

Author Response

Materials MDPI

July 05, 2023

Subject: Response to comments on paper ID: materials-2456064

We thank the reviewers for the valuable comments about our article, which helped us to improve it. So, we are pleased to re-submit for publication the revised version of our paper entitled “Photocatalytic Evaluation and Application as a Sensor for Toxic Atmospheres (Propane and Carbon Monoxide) of Nickel Antimonate (NiSb2O6) Powders”, which contains corrections according to all reviewer’s remarks. We hope the manuscript’s updated version meets your expectations.

 Reviewer 4

Comments and Suggestions for Authors

 The article describes the method of obtaining the material, provides its characterization and the results of the study of sensory properties. The synthesis is well described and the characterization is well done. A few notes:

Comment 1: It is unclear why antimony chloride rather than nitrate was chosen as the precursor. The use of chlorides in the production of sensory materials tends to be avoided, since they remain in the final material and contribute significantly to its electrical conductivity, which reduces the sensory response. Nitrates thermally decompose, so they are preferred for obtaining sensory materials.

Response: It has been documented that by implementing chlorides as precursor materials, a better crystalline phase can be obtained compared to other methods [11, 12, 14, 28, 29]. In addition, from 600 °C, secondary phases associated with chlorine are eliminated from the material. This can be corroborated by the diffractogram in Figure 3 of our manuscript.

Comment 2: In sensory journals, authors usually show "raw", raw data that shows the drift of the background resistance of the sample, the time to reach stationary values ​​after the injection of the target gas, the relaxation time after replacing the target gas with air. The article would have won with such data.

Response: We agree. Details about the measurement process in the sensing system were added (see lines 127-133).

Comment 3: In scientific research, not only the result itself is important, but also the number of significant figures in this result, which indicate the accuracy of the method. The authors present the size of nanorods with an accuracy of three significant figures (77.0 nm), however, electron microscopy methods do not allow one to determine the size with such accuracy - two significant figures would be enough. It is even more strange to see the value of the standard deviation, for which four significant digits of 34.82 nm are given, although the standard deviation should contain as many digits after the decimal point as the value whose error it characterizes.

Response: Absolutely. The data has been set to 2 significant digits.

Comment 4: A similar remark should be made about the results of sensory response. First, it is incorrect to refer to sensory response as sensitivity in Fig. 9, 10 (this error, unfortunately, occurs in many papers). Sensitivity is the derivative of the response with respect to concentration. Secondly, four significant digits for a sensory response is unreasonably large, since both the background resistance value always drifts, and the final resistance value does not remain stationary after the application of the target analyte. Two significant digits would suffice. In some cases, three significant figures can be used, but this needs justification.

Response: We agree and have replaced sensitivity with gas sensing response and reduced the sensing response values to 2 significant digits.

Sincerely,

Corresponding author: Dr. Jacob Morales Bautista

Professor - Research Scientist, Department of Engineering Projects, CUCEI

University of Guadalajara

Blvd. Marcelino García Barragán #1421, 44430 Guadalajara, Jalisco, México.

+52 (33) 1378 5900 ext. 27655

jacob.mbautista@academicos.udg.mx

Round 2

Reviewer 3 Report

The author has properly corrected the manuscript in the revised version , so I recommend that the manuscript can be accepted in the present form